# Vitamin D, the Sunshine Molecule That Makes Us Strong: What Does Its Current Global Deficiency Imply?

**DOI:** 10.3390/nu16132015

**Published:** 2024-06-26

**Authors:** Paolo Riccio

**Affiliations:** Independent Researcher, 70126 Bari, Italy; paoloxriccio@gmail.com

**Keywords:** vitamin D_3_, vitamin D_2_, vitamin D deficiency, vitamin D insufficiency, vitamin D pandemic, vitamin D assay, vitamin D supplementation, vitamin D as disease marker, VDR, chronic inflammatory diseases

## Abstract

Vitamin D_3_ deficiency and insufficiency are becoming a common global issue for us, especially in the most industrially developed countries. The only acknowledged activity of vitamin D_3_ in vertebrates is to promote the absorption of calcium and, therefore, allow for the mineralization of bones. Accordingly, its deficiency is associated with diseases such as rickets. Other numerous vital functions associated with vitamin D_3_ are yet to be considered, and the function of vitamin D_2_ in plants is unknown. Thus, 100 years after its discovery, the importance of vitamin D still seems to be unacknowledged (except for rickets), with little attention given to its decrease throughout the world. In this review, I suggest that vitamin D deficiency and insufficiency may be linked to the westernized lifestyle in more developed countries. Furthermore, I suggest that, rather than the calcemic activity, the main function of vitamin D is, in general, that of strengthening living organisms. I conclude with the hypothesis that vitamin D deficiency may represent a marker for a greater risk of chronic inflammatory diseases and a shorter life expectancy.

## 1. Introduction

A large part of the world’s population, regardless of latitude, has serum vitamin D values below the reference ranges [1]. Depending on the reference range used, this decrease is considered to be deficiency or insufficiency. This trend of decreasing vitamin D perhaps began with the industrial revolution and has been documented for at least the last thirty years.

The prevalence of vitamin D deficiency and insufficiency is so widespread that it can be considered to be a pandemic [2], but it is currently neglected or played down by government institutions: physicians do not often request vitamin D assays, and vitamin D supplementation is usually insufficient and therefore ineffective in the amount administered.

A century after its discovery [3,4], the importance of vitamin D for our health still seems to be unacknowledged. Therefore, the issue is whether vitamin D deficiency is a pandemic that needs, or not, to be taken into account and counteracted, and why the problem should be addressed. Is it sufficient just to avoid getting rickets or can it be helpful to avoid other, perhaps more lethal, diseases?

In fact, we must consider that the active form of vitamin D_3_ behaves like a hormone by binding to a nuclear receptor/transcription factor, called VDR, which has numerous pleiotropic effects at the level of the immune, cardiovascular, neurological, and intestinal systems [5,6,7]. These effects, which have not yet been taken into due consideration by government institutions, are often associated with the inflammatory diseases of our time. In fact, with the increase in vitamin D deficiency, an increase in chronic inflammatory diseases, rather than rickets, occurs.

A possible hypothesis is that there may be a close relationship among the current western lifestyle in more developed countries, multifactorial chronic inflammatory diseases, and vitamin D deficiency.

## 2. Vitamin D

### 2.1. Vitamin D, the Sunshine Molecule That Makes Us Strong

The vitamin D system is very ancient. It has been hypothesized that vitamin D was synthesized in the form of vitamin D_2_ (Ergocalciferol) in photosynthetic eukaryotes as early as one and a half billion years ago [3]. The UV absorption of vitamin D_2_ precursors was probably useful for protecting the photosynthetic eukaryotes from solar radiation, but, at the same time, this process served to open channels for calcium, favoring its absorption.

Nowadays, this mechanism of calcium absorption, now exerted in our organisms by vitamin D_3_, is still in our metabolism and is considered to be important at the intestine level for the absorption of calcium and its subsequent use in bone metabolism, preventing or treating rickets and giving strength and stability to our bone structure. However, calcium is not only important for the formation of bones and teeth, but it is also a strong activator of cellular metabolism, so much so that its concentration in the cell is kept low (10^−7^ M), approximately four orders of magnitude lower than that of extracellular calcium (10^−3^ M). The intracellular concentration of calcium is finely regulated and kept at low levels through the following processes: 99% of calcium is bound to cellular components or is sequestered both in the endoplasmic reticulum and in mitochondria, while free calcium is released from the cell by Ca-ATPase and Na–Ca exchange. Thanks to sunshine, which allows its synthesis, vitamin D makes us stronger both structurally and metabolically, since it also has a favorable role for numerous other functions.

Vitamin D is the only vitamin/hormone that originates from sunshine. We can assume that the production of both vitamin D_2_ in fungi and yeasts and vitamin D_3_ in humans and animals serves to support life on earth, not as chemical energy, like ATP, but as a link between solar energy and the correct functioning of living organisms. The role of vitamin D in the formation and maintenance of a solid bone structure can be extended to our entire body. Thus, vitamin D can be seen as a molecule that makes us strong.

### 2.2. The Vitamin D Receptor (VDR)

Usually, when it comes to vitamin D, only its role in calcium absorption and bone metabolism is taken into consideration, and, therefore, its anti-rickets action. However, in almost all human cells there is a vitamin D receptor (VDR) whose functions are not correlated with skeletal action alone, but also coincide with those of a transcription factor [8,9]. The existence of a receptor fits well with the idea that calcitriol, the active form of vitamin D, acts as a hormone, a hormone that, in my opinion, supports us in our activities and protects us from numerous inflammatory diseases.

Numerous studies strongly suggest that, in accordance with the fact that the VDR receptor is widespread across almost all our cells, vitamin D has many extra-skeletal effects such as the regulation of cell proliferation, immune and muscle function, skin differentiation and reproduction, vascular and metabolic properties, as well as growth and development. These findings explain why vitamin D deficiency, besides enhancing the risk of osteoporosis and fractures, is associated with many diseases [6].

### 2.3. The Characters in This Story

The characters in this story are (1) cholecalciferol, which is the natural form of vitamin D_3_, produced by the action of sunlight on animal skin from 7-dehydrocholesterol, (2) calcidiol (or calcifediol), 25-hydroxyvitamin D_3_, the inactive circulating metabolite of vitamin D which is synthesized in the liver and is the precursor of calcitriol, and (3) calcitriol [1,25-dihydroxyvitamin D_3_, the active form of vitamin D_3_], which is synthesized in the kidney and considered to act like a hormone, while calcidiol is considered to be a pre-hormone. Calcidiol is the form of vitamin that is measured in serum and referred to when talking about deficiency or insufficiency.

Thus, when we talk about vitamin D, we mean a system composed of the three forms mentioned above: cholecalciferol (vitamin D_3_) as the form of vitamin D synthesized following exposure to sunlight, calcidiol [25(OH)D] as the circulating form of vitamin D of which the dosage is made, and calcitriol [1,25(OH)_2_D] as the active form [10].

### 2.4. Vitamin D_3_ Metabolism [5]

As already mentioned above, the formation of vitamin D_3_ (cholecalciferol) occurs in the skin of animal organisms exposed to sunshine through the action of UV light on 7-dehydrocholesterol (Figure 1). The animal’s diet is the other source of vitamin D that, while less important than sunshine itself, is also dependent on sunshine (Figure 1). The intake of vitamin D_3_ in the diet usually corresponds to about 10% of that produced following exposure to the sun and is, therefore, much less important. Vitamin D is transported to the liver by the vitamin-D-binding protein.

From here on, the metabolism of vitamin D_3_ depends on four cytochrome P450 (CYP) enzymes. Briefly, in the liver, the mitochondrial CYP27A1 or the microsomial CYP2R1 convert cholecalciferol into calcifediol (calcidiol) [25(OH)D_3_] by hydroxylation in position 25. The synthesis of calcidiol is inhibited by calcidiol itself.

In the kidney, the enzyme CYP27B1 converts 25(OH)D_3_ in 1–25(OH)_2_D_3_, i.e., calcitriol, the active form of vitamin D. Inhibitors of CYP27B1 are the inhibitors of tyrosine kinase, used in clinical oncology for the treatment of some malignancies.

Calcitriol has a half-life of about 12 h, being inactivated by the mitochondrial CYP24A1. CYP24A1 is positively modulated by the same substrate calcitriol. Thus, the concentration of 1,25 dihydroxyvitamin D, i.e., calcitriol, cannot increase if the enzyme is active.

Low serum concentrations of vitamin D_3_ (calcidiol) may result from the inhibition of CYP enzymes involved in the synthesis and/or from the increased activity of CYP24A1 enzyme in the catabolic process.

Exposure of almost the entire body to sunlight for an entire day can produce as much as 10,000 IU (International Units) of cholecalciferol, i.e., approximately 250 IU of 25(OH)D_3_, i.e., calcidiol [10]. Of these, only about 3% (2 IU) will become 1,25(OH)_2_D_3_, the active form. This means that the intake of even 10,000 IU vitamin D per day cannot be toxic, since the presence of the active form remains rather low. Furthermore, only the increase in calcitriol, the active form, can cause hypercalcemia in some individuals.

It is also interesting to consider that, although the half-life of calcitriol is only 12 h, the reserves of cholecalciferol in the adipose and muscle tissues can allow its synthesis for about a couple of months [10].

### 2.5. Vitamin D—Why It Is Present in Plants and Invertebrates

It is generally believed that vitamin D_2_, i.e., ergocalciferol, is the form of vitamin D found in the plant world. It is formed from ergosterol through the action of sunlight. Dietary vitamin D_2_ is found especially in mushrooms, beans, and dried fruit. The plant vitamin D_2_ is analogous to the vitamin D_3_ in vertebrates and is recognized by enzymes that have cholecalciferol as a substrate. Thus, we can utilize both vitamin D_2_ and vitamin D_3_, although vitamin D_2_ is less effective.

However, there is no satisfying information regarding the production and the role of vitamin D_2_ in plants, and the exact functions of vitamin D_2_ in the plant kingdom still remain to be elucidated [11,12]. Instead, most articles report the effects of vitamin D_2_ on humans.

Although plants have a complex sterol mixture in addition to ergosterol [12], they do not produce vitamin D on their own. The only exception is the Solanaceae family that produce vitamin D_3_ [12]. The production of vitamin D_2_ only occurs in mushrooms and yeasts. Plants containing endophytic fungi as contaminants may contain vitamin D_2_, but only because of fungal contamination [13].

Regarding the function of vitamin D in plants, since it seems that they do not have either vitamin-D-binding proteins or receptors but still have calcium channels and pumps like animals, it can be assumed that, in plant cells, vitamin D intervenes in the calcium metabolism. Thus, in plants, calcium should be required for growth stimulation, root initiation, and germination [14].

Besides plants, production of vitamin D also occurs in invertebrates such as various insect species and worms [15]. The question arises as to why this occurs.

All this would suggest a multifaceted role played by vitamin D in humans that extends beyond calcium regulation and involves various physiological extraskeletal processes.

The major function of vitamin D_3_ in humans is believed to be the maintenance of calcium homeostasis. Its insufficiency increases the risk of osteoporosis, as well as that of numerous diseases such as preeclampsia, childhood dental caries, periodontitis, infectious diseases, hypertension, cardiovascular disease, autoimmune diseases, type 2 diabetes, neurological disorders, and cancer [2,16]. The question then arises as to why vitamin D is present in the plant world and in invertebrates. Apparently, there are not many studies on the functions of vitamin D in plants, as reports on vitamin D_2_ usually deal with the effects on humans [13]. However, it may be that the function of vitamin D_2_ is to absorb calcium into the stems and, more generally, it may serve as a means of making plants stronger and more resistant to adverse conditions, as vitamin D_3_ probably does in animal organisms. Obviously, we cannot assume a role of vitamin D_2_ in plants and invertebrates that concerns a skeletal apparatus.

### 2.6. Assessment of Vitamin D and Reference Ranges

When we talk about vitamin D levels, we are referring to the measurement of 25(OH)D in serum. Its values are expressed either as nmol/L, ng/mL, or µg/mL. A serum level of 120 nmol/L (48 ng/mL) corresponds to the value naturally occurring after regular sun exposure [17]. The most used reference ranges have a sufficiency cut-off of 50 nmol/L (20 ng/mL) [NIH, National Institute of Health, and IOM, Institute of Medicine, now NAM, National Academy of Medicine] or 75 nmol/L (30 ng/mL) (ES, Endocrine Society US, Washington, DC, USA). For ES, the acceptable concentration range is 75–250 nmol/L (30–100 ng/mL), preferably 40–60 ng/mL, and a third cut-off was set at 32 ng/mL [18].

It is likely that higher levels of vitamin D are necessary for extraskeletal functions mediated by VDR. However, in some individuals, higher levels of vitamin D may lead to a greater absorption of calcium and this may increase the risk of coronary heart disease. In these cases, to have a complete picture of the clinical situation, the analysis of serum 25(OH)D levels can be completed by using calcium, phosphate, alkaline phosphatase, and parathyroid hormone (PTH).

### 2.7. Administration of Vitamin D

Vitamin D supplements are indicated mainly in the form of International Units (IU). Even in this case, there is no consensus on the quantities to be administered. The Institute of Medicine (IOM) and the European Food Safety Authority (EFA) recommend an intake of 600 IU/d, while the UK Scientific Advisory Committee on Nutrition (SACN) recommends 400 IU/d, and the World Health Organization recommends 200 IU/d. The Endocrine Society (ES) recommends 1500–2000 IU/d for adult people and 2–3 times more for obese adults. The safety limit in the administration of Vitamin D is 10,000 IU/day according to the ES, but only 4000 IU/day according to the NAM [17]. I personally recommend the administration of 5000 IU/d (50,000 IU/10 days or 100,000 IU/20 days). The administration of 5000 IU/d, lower than that which would occur after a day of sun exposure, is just higher than that proposed by the National Academy of Medicine (NAM), i.e., 4000 IU/d. and has been used in many other studies [19].

### 2.8. Vitamin D in Pregnancy

There are several conditions to take into consideration when we want to evaluate the benefits of vitamin D beyond the skeletal effects, for example, old age or the various chronic inflammatory diseases of our time, but the most interesting and important condition is undoubtedly pregnancy, because it concerns two individuals: a human organism in the first months of growth and the mother. The mother can encounter complications that could perhaps be controlled by the sufficient presence of vitamin D, but, if vitamin D is insufficient, there can be complications for the fetus that affect its whole life. Vitamin D is particularly important in the early stages of life.

In fact, it has been observed that, during fetal development, the concentration of calcidiol (25(OH)D) increases, so there is a natural demand for this vitamin in pregnancy [20], certainly for the development of the fetus’s bone system but also, more generally, for growth.

With regard to complications, in two studies, it has been found that vitamin D deficiency increases the risk of preeclampsia by 65% and constitutes a risk factor for gestational diabetes [21]. Two other studies have revealed that concentrations above 40 ng/mL of 25(OH)D in maternal serum reduce the risk of preterm birth by 59% [22], while sufficient concentrations of vitamin D reduce the risk of autism by 58% [23].

### 2.9. Low Levels of Vitamin D Could Be a Marker of Potential Predisposition to Multifactorial Chronic Inflammatory Diseases

That of vitamin D is perhaps the only case in which there are two main different reference ranges for the assay in serum: that of the IOM (NAM) with a cut-off value of 20 ng/mL (50 nmol/L) and that of the ES with a cut-off value of 30 ng/mL (75 nmol/L). The value of 30 ng/mL corresponds to the minimum plateau value of PTH and, therefore, should be fine with regard to the skeletal action of vitamin D. However, there are extraskeletal effects of vitamin D. These non-skeletal effects occur at concentrations higher than 30 ng/mL, probably between 40 ng/mL and 60 ng/mL [20]. It has been reported that concentrations above 50 ng/mL can increase the expression of VDR mRNA [9] and that calcitriol binds to VDR with a K_d_ of 0.1 nM [7]. So, at higher concentrations, vitamin D binds to its receptor and forms a heterodimer with the vitamin A receptor, exerting its anti-inflammatory action. The anti-inflammatory action is anti-aging, modulates the innate immune response, and counteracts the onset of inflammatory diseases. This means that low vitamin D values, including those around 20 ng/mL, may be unsafe and could be considered to be a sign of risk of chronic inflammatory diseases.

Table 1 has been compiled from research on PubMed into the correlation between vitamin D deficiency and possibly associated diseases, as well as on their prevalence rate. Although the analysis of the number of published articles from PubMed is only indicative, the data reported in Table 1 make it clear that the higher number of studies on vitamin D deficiency and rickets is probably due to the fact that vitamin D deficiency is mistakenly thought to be mainly associated with rickets, especially because it does not correspond to its prevalence rate. Table 1 also shows that the prevalence of vitamin D deficiency, cancer, cardiovascular diseases, and chronic inflammatory diseases have all been increasing in recent years, at least in terms of the number of publications, so it is reasonable to assume the existence of a correlation among them.

The data in Table 1, Section 2, are in agreement with health organizations, such as the World Health Organization (WHO) and the Global Burden of Disease Study (GBD), which have reported on the rising rates of cancer and chronic inflammatory diseases in recent years.

In particular, GBD has identified air pollution as the first among 88 leading risk factors that can result in diseases in individuals that are exposed to it [24]. In this exhaustive report, it is also shown that, amongst others, important risk factors include diets rich in trans-fat, sodium, and alcohol and low in vegetables, fruit, and whole grains.

In my interpretation, air pollution could contribute to vitamin D deficiency, whereas the western diet and lifestyle could contribute to the increase in both vitamin D deficiency and chronic inflammation. These would be the risk factors that link the decrease in serum vitamin D with the increase in chronic inflammatory diseases and cancer shown in Table 1.

Accordingly, as shown in Figure 2, the decrease in serum 25(OH)D_3_ levels can be attributed to several factors: (1) those decreasing the exposure to sunlight, e.g., air pollution, latitude, winter season, the use of sunscreen, and indoor activities, (2) those related to lifestyle, such as a high-calorie diet and a sedentary lifestyle with a consequent increase in fat deposits, and (3) environmental factors that inhibit the synthesis or upgrade the degradation of calcitriol [1,25 (OH) D_3_], examples of which include pesticides in agriculture, additives in industrial food, or medical drugs.

### 2.10. What to Do in the Future to Counteract the Prevalence of Vitamin D Deficiency and Insufficiency: Some Suggestions

First, a general screening of the population should be carried out. If there is no initiative at a government level, everyone should do it at a personal level.

The first dosage, in addition to that of calcidiol, should also include a dosage of calcium, phosphate, and PTH. The analysis should be redone at least once a year, after a two-month break in the summer period.

Given the range of 40–60 ng/mL as a reference, if you are below these values, you need to proceed with supplementation. This should correspond to the intake of approximately 5000 IU per day, i.e., 50,000 IU every 10 days or 100,000 IU every 20 days, always under medical supervision.

I personally recommend taking vitamin D together with vitamin A [25] (around 2000–3000 IU/d, however no more than 9000 IU/d, with 10,000 IU being the maximum recommended limit) [19].

## 3. Discussion

### Doubts and Contradictions Regarding Vitamin D

After its discovery a hundred years ago and a huge number of studies, we can ask ourselves whether we really know this vitamin well.

First of all, is vitamin D a vitamin or a hormone? [10,26] The answer is: it can be both. We can speculate that it is a vitamin because (1) it cannot be synthesized and must be introduced from the outside (in this case formed through the action of sunlight), (2) vitamin D deficiency, as for other vitamins, causes specific diseases, in this case rickets in children and osteomalacia in adults, and (3) the reactions of vitamin D are of first order (depend on the concentration of the substrate), while the reactions of the formation of steroid hormones are of zero order (they are independent of the concentration of the substrate) [10]. On the other hand, calcitriol is a hormone as it derives from a derivative of cholesterol and its metabolism requires CYP 450 enzymes, as is the case with steroid hormones.

There is some confusion at various levels, starting with terminology and roles of vitamin D. We call vitamin D_3_ cholecalciferol, the molecule that is formed in the skin thanks to sunlight, we measure the circulating 5-hydroxyvitamin D, which is formed in the liver and which is inactive. We also call it vitamin D, just as we also call the active form, calcitriol, vitamin D, the 1,25 dihydroxyvitamin D which is formed prevalently in the kidney.

There is no agreement either on the need to monitor vitamin D levels in the serum or on what the reference values should be, as well as on the quantities of vitamin D to be administered per day. Often, the results of RCTs regarding the effects of vitamin D on chronic diseases remain elusive because vitamin D administration is too low to see its effects.

Yet there is no consensus on which should be considered the reference range in vitamin D assays: those with a 20 ng/mL cut-off value or those with a 30 ng/mL cut-off value?

Like the plant world with its photosynthesis, the animal world is also capable of directly using sunlight, even if we generally do not think about it. The aim is always the same: to obtain energy. Except that for animals it is not a question of converting solar energy into chemical energy, but it is a much less direct way of transferring energy through the formation of the vitamin D molecule. Vitamin D converts solar energy into vital energy at different levels.

Vitamin D_3_ is known for its effects on calcium and phosphate metabolism and, therefore, for its skeletal effects. However, mushrooms and a few plants also produce vitamin D in the form of vitamin D_2_, which is obviously not there for its skeletal effects. What comes from solar energy must serve as a form of energy: a support for living organisms. In my opinion, this should be the logic of the formation of vitamin D in living organisms.

It has been reported that vitamin D has many extraskeletal effects: anti-aging, the favoring of intestinal eubiosis, anti-inflammatory action, action against chronic inflammatory diseases, cancer, and mortality [27,28,29]. However, these effects can occur at concentrations higher than those recommended by the IOM, i.e., 40–60 ng/mL, necessary for the binding to the VDR [9,25].

However, the current decrease of serum vitamin D does not appear to be related to the number of cases of rickets and osteomalacia in the same country, while the spread of chronic inflammatory diseases and cancer, apparently still unrelated to vitamin D deficiency, is increasing.

If we want the protective action of vitamin D in humans to be safeguarded, we must consider that vitamin D deficiency is indeed a problem and it is of a pandemic nature. It is necessary to intervene in order to have higher levels of vitamin D. Since vitamin D values are low even in countries with a lot of sunshine, one might think that exposure to sunlight is not enough, and that adequate administration of vitamin D must be provided. The supplementation cannot be just 200–600 IU/day as suggested by WHO (World Health Organization), SACN (UK Scientific Advisory Committee on Nutrition), and EFA (European Food Safety Authority) [17], but it must be between 4000 IU/day and 5000 IU/day in order to reach the range of 40–60 ng/mL.

In one of our pilot studies with multiple sclerosis patients, the administration of almost one thousand IU vitamin D_3_/day was found to be insufficient, and vitamin D levels continued to slowly decline [30]. Since the IU/day of vitamin D suggested for supplementation are probably insufficient, it is possible that some interventional trials might have not been able to clearly highlight the extraskeletal roles of vitamin D. It cannot be said that there is no confirmation of vitamin D effects in RCTs if the quantities of vitamin D administered are not increased to 4000–5000 IU/day.

There is no need to be afraid of administering doses of vitamin D of 5000 IU. Adverse effects are rare when vitamin D supplements are given, also because there is self-regulation: calcidiol inhibits its synthesis, calcitriol increases its degradation. Moreover, no adverse effects have been observed even at 10,000 IU/d [31].

## 4. Conclusions

It is time to have a broader vision of the role of vitamin D in our health. Active vitamin D acts as a hormone through its receptor VDR, a nuclear receptor/transcription factor found in almost all our cells. Already from this wide distribution, it can be deduced that its function cannot be only to control calcium homeostasis and bone formation: vitamin D shows multiple mechanisms of action and pleiotropic effects, suggesting that its role may be that of a general support to our activities which include immune defense, cardiovascular function, brain activity, and intestinal microbiota health. This role has a preventive rather than therapeutic nature. Accordingly, a strong correlation has been found between the prevalence of severe vitamin D deficiency and population mortality rate from COVID-19 in Europe [32].

Vitamin D is the only vitamin that originates from sunshine. It can be seen as a vital energy source that supports and harmonizes the most diverse functions of our organism, thus supporting human health and not just playing a role related to calcium and bone metabolism.

There are now numerous indications that vitamin D deficiency is associated with many chronic inflammatory diseases in humans. It may be that its deficiency is due to the same multifactorial causes that facilitate the onset of chronic inflammatory diseases, for example, adiposity, junk food, food additives in processed food, indoor sedentary lifestyle, reduced sun exposure, abuse of medicines, pesticides, and stress. Could vitamin D deficiency be a marker of susceptibility to such diseases and of a shorter life expectancy? This is a hypothesis to work on.

## Figures and Tables

**Figure 1 nutrients-16-02015-f001:**
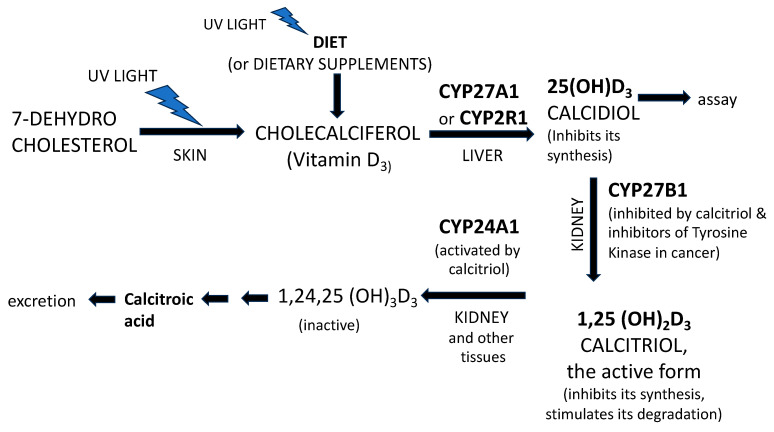
Schematic representation of vitamin D metabolism.

**Figure 2 nutrients-16-02015-f002:**
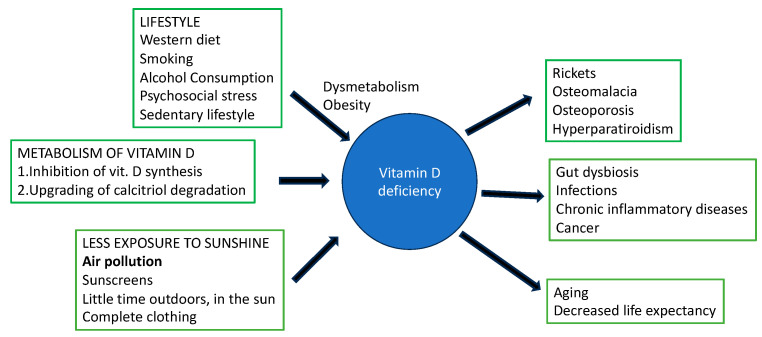
Relatioship between lifestyle, vitamin D deficiency and human chronic diseases as well as some unhealthy conditions.

**Table 1 nutrients-16-02015-t001:** PubMed search on the correlation between vitamin D deficiency and possibly related diseases (section 1), as well as on their prevalence rate (section 2).

PUBMED Search	Results (No Filters) (A)	Results 1983–2003 (B)	Results 2004–2024 (C)	Fold Increase 2004–2024/ 1983–2003
Section 1				
1.1. Vitamin D deficiency & Rickets	15,288	4006	4809	1.2
1.2. Vitamin D deficiency & chronic inflammatory diseases	462	22	440	20.0
1.3. Vitamin D deficiency & cardiovascular disease	3291	165	2905	17.6
1.4. Vitamin D deficiency & cancer	4035	452	3215	7.1
Section 2				
2.1. Vitamin D deficiency prevalence rate		45	1072	23.8
2.2. Rickets prevalence rate		44	122	2.8
2.3. Chronic Inflammatory Diseases prevalence rate		288	2281	7.9
2.4. Cardiovascular disease prevalence rate		27,267	94,337	3.5
2.5. Cancer prevalence rate		47,470	135,244	2.9

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
