# Peer review of "Vitamin D, the Sunshine Molecule That Makes Us Strong: What Does Its Current Global Deficiency Imply?"

_nutrients, 2024, doi:10.3390/nu16132015_

Round 1

Reviewer 1 Report

Comments and Suggestions for Authors

With great interest I read the review article on vitamin D from colleague Paolo Riccio. Raising awareness around vitamin D and the increasing rates of deficiency and insufficiency is important. However, I do have a few remarks, mainly on the strongness of the wording used to assume causality. There are too many statements throughout the text without data/references to back it up to call this article a review.

-          I am not aware that the function of vitamin D2 in plants is unknown – to the best of my knowledge D2 and D3 are the same vitamin with the same function, but D2 has plant origin and D3 has animal origin. As stated later in the article, the potency of D2 might be less, but it still is of value and most assays measure both D2 and D3.

-          I don’t think there is enough evidence to suggest that ‘rather than the calcemic activity, the main function of vitamin D is in general that of strengthening living organisms’. There are numerous studies on the extra-skeletal benefits of vitamin D, but it is an overinterpretation of the data to say it is ‘strengthening living organisms’.

-          Also, the conclusion ‘that vitamin D deficiency may represent a marker of greater risk of chronic inflammatory diseases and shorter life expectancy’ is too strong – I would suggest rewording this into a hypothesis.

-          The causality suggested in this sentence is too much: ‘In fact, with the increase in vitamin D deficiency we have more of an increase in chronic inflammatory diseases, rather than in rickets’. Moreover, saying ‘vitamin D makes us stronger both structurally and metabolically, since it also has a favorable role for cardiovascular, immune, nervous and enteric functions. Vitamin D supports us everywhere and in all our activities’ without a reference/data to back this up, it is again an overinterpretation of available data.

Comments on the Quality of English Language

The English needs some editing too, as there are a few ‘non-English’ sentences structures used (‘vitamin D assay is often not prescribed’ and ‘2. This theme is devloped in this review.’). Use of ‘In conclusion’ in the middle of the article is inappropriate.

Author Response

RE to Reviewer 1

  1. I do have a few remarks, mainly on the strongness of the wording used to assume causality. There are too many statements throughout the text without data/references to back it up to call this article a review.

I obviously agree with the reviewer and I thank him for his comments.

In the revised version I try to make less drastic statements and to present some of them in the form of hypotheses, but as the reviewer himself states, we need to have much more awareness of the importance of vitamin D for human health, not only with regards to rickets and osteomalacia. We also need to take action on the prevalence of vitamin D deficiency or insufficiency and we need to motivate our interventions beyond the effects of vitamin D on our bone structure.

  1. I am not aware that the function of vitamin D2 in plants is unknown – to the best of my knowledge D2 and D3 are the same vitamin with the same function, but D2 has plant origin and D3 has animal origin.

As much as I investigated, I was unable to find satisfying information regarding the role of vitamin D in plants. Probably, the exact functions of vitamin D in the plant kingdom still  remain to be elucitated. Instead, all the articles referenced the effects of vitamin D2 on humans.

From what I understand, the production of vitamin D2 occurs only in mushrooms and yeasts. Plants containing endophytic fungi as contaminants may contain vitamin D2, but only because the fungal contamination.

All things considered, studies on the production of vitamin D in plants are few and do not yet give clear results. In some cases the results are contradictory and difficult to evaluate. The synthesis of vitamin D2 in plants remains unclear (Li, Y., Yang, C., Ahmad, H., Maher, M., Fang, C., and Luo, J. (2021). Benefiting others and self: Production of vitamins in plants. J. Integr. Plant Biol. 63: 210–227).

Regarding the function of vitamin D in plants, since it seems that they do not have both vitamin D binding proteins and receptors but still have calcium channels and pumps like animals, it can be assumed that vitamin D intervenes in the metabolism of calcium in the physiology of the plant cell. Thus, in plants, calcium should be required for growth stimulation, root initiation and germination (Dodd, AN, Kudla J, Sanders D. The language of calcium signaling. Ann. Rev. Plant Biol. 2010, 6, 593- 620.)

Besides plants, production of vitamin D is also present in invertebrates such as various insect species and worms. The question arises as to why this occurs.

Overall, we may suggest a multifaceted role for vitamin D that extends beyond calcium regulation and involves various physiological extra-skeletal processes.

  1. I don’t think there is enough evidence to suggest that ‘rather than the calcemic activity, the main function of vitamin D is in general that of strengthening living organisms’. There are numerous studies on the extra-skeletal benefits of vitamin D, but it is an overinterpretation of the data to say it is ‘strengthening living organisms’.

Studies published after peer reviews on the non-calcemic role of vitamin D are now very numerous. In these studies on the extraskeletal effects of vitamin D it was found that vitamin D supports various functions of the human organism, at the  immunological, neurological, enteric, and cardiovascular levels including growth and development.

Looking with a unitary vision at the extraskeletal effects of vitamin D, it can be suggested that vitamin D has the general function of supporting our organism and therefore it can be hypothesized that it makes us stronger and more resistant to infections, chronic inflammatory diseases and cancer. However, these effects can be achieved only at higher vitamin D concentrations, i.e. between 40 and 60 ng/mL. This is important.

  1. Also, the conclusion ‘that vitamin D deficiency may represent a marker of greater risk of chronic inflammatory diseases and shorter life expectancy’ is too strong – I would suggest rewording this into a hypothesis.

I agree. Done.

  1. The causality suggested in this sentence is too much: ‘In fact, with the increase in vitamin D deficiency we have more of an increase in chronic inflammatory diseases, rather than in rickets’. Moreover, saying ‘vitamin D makes us stronger both structurally and metabolically, since it also has a favorable role for cardiovascular, immune, nervous and enteric functions. Vitamin D supports us everywhere and in all our activities’ without a reference/data to back this up, it is again an overinterpretation of available data.

I agree and I am more cautious in the revised version.

However, I would like to point out that in the revised work I include data obtained from PubMed showing that there are more studies on the correlation between vitamin D deficiency and rickets than on the correlation between vitamin D deficiency and chronic inflammatory diseases and cancer, but it is the latter that are increasing just as vitamin D deficiency is increasing.

Furthermore, our awareness of the increase in diseases such as chronic inflammatory diseases and cancer compared to rickets already derives in some way from our life experience. Furthermore, as already mentioned, numerous publications indicate that the prevalence rate of cancer, cardiovascular and chronic inflammatory diseases is increasing. This upward trend is confirmed by international health organizations such as the World Health Organization (WHO) [https://www.who.int/news-room/fact-sheets/detail/noncommunicable-diseases;   https://www.who.int/publications; https://www.who.int/news/item/01-02-2024-global-cancer-burden-growing--amidst-mounting-need-for-services] and the Global Burden of Disease Study (https://www.healthdata.org/research-analysis/gbd; https://ghdx.healthdata.org/gbd-2021).

  1. Comments on the Quality of English Language

The English needs some editing too, as there are a few ‘non-English’ sentences structures used (‘vitamin D assay is often not prescribed’ and ‘2. This theme is devloped in this review.’). Use of ‘In conclusion’ in the middle of the article is inappropriate.

The two short sentences quoted were like two notes added by myself after the revision of the English by a native-speaker translator from the University of Bari. I have deleted one of them.  In any case, the revised version has been edited by the native-speaker Richard Lusardi.

Reviewer 2 Report

Comments and Suggestions for Authors Dear author, the review paper “VITAMIN D, THE SUNSHINE MOLECULE THAT MAKES US STRONG: WHAT DOES ITS CURRENT GLOBAL DEFICIENCY IMPLY?” present interesting data of vitamin D and discuss important aspects of Vitamin D intake/uptake, effect and consequence of deficiency. In my perspective, some reference should be add/revised. Here are my comments.
Introduction.

 The author writes “In fact, with the increase in vitamin D deficiency we have more of an increase in chronic inflammatory diseases, rather than in rickets”. What is the literature evidence of this fact?

The same of comments for the next paragraph. “There is a close relationship between the current Western lifestyle in more developed countries, multifactorial chronic inflammatory diseases, and vitamin D deficiency”

-          Point 2.1.

For example, the author writes “Vitamin D is the only vitamin/hormone…”. Ellison and Moran published an interesting paper about this topic. Nurs. Clin. North Am. 2021 Mar;56(1):47-57. doi: 10.1016/j.cnur.2020.10.004

-          Point 2.3.

Please correct: 1,25-hydroxy-vitamin D3 to 1,25-dihydroxyvitamin D3

-          Point 2.4.

Please clarify/correct: 1,25 not 1-25/ 24,25 not 24-25.

Figure 1: What is PTH and IU? Note that at Point 2.6 and 2.7 the author elucidates these abbreviations, it should appear at Point 2.4.  

In the figure 1 appear “ assay”. It is correct?

-          Point 2.7.

The author says “I personally recommend the administration of 5000 IU/d (50000 IU/10 days or 100000 IU/20 days).” Why? There are some evidences for this recommendation?

-          Point 2.8

“…preeclampsia by 65% and constitutes a risk factor for gestational diabetes [14].” This reference should be replaced by the direct paper that this information was found. “Yuan Y, Tai W, Xu P, Fu Z, Wang X, Long W, et al. Association of maternal serum 25-hydroxyvitamin D concentrations with risk of preeclampsia: a nested case-control study and meta-analysis. J Matern Fetal Neonatal Med. 2019:1–10.

-  Point 2.9.

It is not clear the role of “dysmetabolism/Obesity” at the Figure 2. Please clarify.
In my opinion, when the author talks about “The use of sunscreen” should explore/explain better. There are a great variety of sunscreen with different: UV index, use specification, effects. There are some evidences that the use of sunscreen UV 50 lead to more Vitamin D deficiency compared to sunscreen UV 30?

Author Response

    RE to Reviewer 2

Dear author, the review paper “VITAMIN D, THE SUNSHINE MOLECULE THAT MAKES US STRONG: WHAT DOES ITS CURRENT GLOBAL DEFICIENCY IMPLY?” present interesting data of vitamin D and discuss important aspects of Vitamin D intake/uptake, effect and consequence of deficiency. In my perspective, some reference should be add/revised. Here are my comments.

Introduction.
o    The author writes “In fact, with the increase in vitamin D deficiency we have more of an increase in chronic inflammatory diseases, rather than in rickets”. What is the literature evidence of this fact?

RE: Numerous publications indicate that the prevalence of cancer and chronic inflammatory diseases is increasing. This upward trend is confirmed by international health organizations such as the World Health Organization (WHO) [https://www.who.int/news-room/fact-sheets/detail/noncommunicable-diseases;   https://www.who.int/publications; https://www.who.int/news/item/01-02-2024-global-cancer-burden-growing--amidst-mounting-need-for-services] and the Global Burden of Disease Study (https://www.healthdata.org/research-analysis/gbd; https://ghdx.healthdata.org/gbd-2021).

In the revised version, I  have now inserted a table showing the literature listed in PubMed possibly indicating that in recent years vitamin D deficiency, chronic inflammatory diseases, cardiovascular diseases and cancer are increasing more than rickets.

o    The same of comments for the next paragraph. “There is a close relationship between the current Western lifestyle in more developed countries, multifactorial chronic inflammatory diseases, and vitamin D deficiency”

RE: We agree. In the revised version this assertion is presented in the form of a hypothesis.

-          Point 2.1.
For example, the author writes “Vitamin D is the only vitamin/hormone…”. Ellison and Moran published an interesting paper about this topic. Nurs. Clin. North Am. 2021 Mar;56(1):47-57. doi: 10.1016/j.cnur.2020.10.004 

RE: Ok. Reference added.

-          Point 2.3.

Please correct: 1,25-hydroxy-vitamin D3 to 1,25-dihydroxyvitamin D3

RE: done.

-          Point 2.4.

Please clarify/correct: 1,25 not 1-25/ 24,25 not 24-25.

RE: done.

Figure 1: What is PTH and IU? Note that at Point 2.6 and 2.7 the author elucidates these abbreviations, it should appear at Point 2.4.  

RE: done

In the figure 1 appear “ →assay”. It is correct?

RE: Yes. I have made the arrow more visible

-          Point 2.7.
The author says “I personally recommend the administration of 5000 IU/d (50000 IU/10 days or 100000 IU/20 days).” Why? There are some evidences for this recommendation?

RE: The administration of 5000 IU/d is for me the dosage required to reach a concentration of 40-60 ng/mL in the serum within a month or so, depending on the starting values. Bear in mind that the maximal daily dose recommended by the NAM is 4000 IU/d. Furthermore, several studies have administered much higher doses without problems, for example in: Amon U, Yaguboglu R, Ennis M, Holick MF, Amon J. Safety Data in Patients with Autoimmune Diseases during Treatment with High Doses of Vitamin D3 According to the "Coimbra Protocol". Nutrients. 2022;14(8):1575.

-          Point 2.8

“…preeclampsia by 65% and constitutes a risk factor for gestational diabetes [14].” This reference should be replaced by the direct paper that this information was found. “Yuan Y, Tai W, Xu P, Fu Z, Wang X, Long W, et al. Association of maternal serum 25-hydroxyvitamin D concentrations with risk of preeclampsia: a nested case-control study and meta-analysis. J Matern Fetal Neonatal Med. 2019:1–10.

RE: Done.

-  Point 2.9.
It is not clear the role of “dysmetabolism/Obesity” at the Figure 2. Please clarify.
In my opinion, when the author talks about “The use of sunscreen” should explore/explain better. There are a great variety of sunscreen with different: UV index, use specification, effects. There are some evidences that the use of sunscreen UV 50 lead to more Vitamin D deficiency compared to sunscreen UV 30?

RE: 
1) Dysmetablism and obesity are two conditions that can result from a lifestyle as indicated before and which can contribute to vitamin D deficiency. 
2) It is generally accepted that anything that limits the absorption of UV light can contribute to vitamin D deficiency - it's more a matter of exposure time.

Round 2

Reviewer 1 Report

Comments and Suggestions for Authors

Thank you for the improvements provided. I still consider this more an opinion article rather than a review. 

Comments on the Quality of English Language

Needs further editing.